# Domain Translation with Monolingual Lexical Distribution

**Yusuke Sakai**                                                    *sakai.yusuke.sr9@is.naist.jp*
*Nara Institute of Science and Technology (NAIST)*
† *Equal Contribution*

**Zhi Qu**                                                          *qu.zhi.pv5@is.naist.jp*
*Nara Institute of Science and Technology (NAIST)*
† *Equal Contribution*

**Hidetaka Kamigaito**                                              *kamigaito.h@is.naist.jp*
*Nara Institute of Science and Technology (NAIST)*

**Taro Watanabe**                                                   *taro@is.naist.jp*
*Nara Institute of Science and Technology (NAIST)*

**Xiaojiang Liu**                                                   *xiaojiang_liu@apple.com*
*Apple*

**Reviewed on OpenReview:** *https://openreview.net/forum?id=UKLBobrFCR*

## Abstract

Neural machine translation (NMT) often demands a large amount of high-quality training data when adapting to a new domain with a carefully designed fine-tuning strategy. However, constructing a sufficient amount of parallel data for training poses challenges even for fine-tuning. This work proposes to fine-tune a generic NMT model using only the monolingual lexical distribution estimated from a small amount of in-domain data in the target language. Word frequency plays a critical role in analyzing the differences among corpora in various fields, e.g., psycholinguistic and language education, and our challenge lies in whether we can fit a model using the naive statistics collected from a target language domain in NMT. We leverage a variant of energy-based models (EBMs) based on Conditional Distributional Policy Gradients (CDPG) with a large number of EBMs to constrain the fine-tuning process with lexical distribution. We conduct experiments across four translation directions and four domain datasets, totaling 16 domain adaptation scenarios. The results demonstrate that our method enables robust domain shift while mitigating catastrophic forgetting, achieving effective domain adaptation using only a small amount of monolingual resources.

## 1 Introduction

Thanks to the crawling technology and corpora construction efforts (Tiedemann, 2012; Bañón et al., 2020; Morishita et al., 2022), we have access to abundant parallel translation data, resulting in the development of high-performance pre-trained neural machine translation (NMT) models. However, NMT models suffer from performance degradation when translating text from the domains different from the domain of the training corpus due to the mismatch of the domain-specific terminologies (Koehn & Knowles, 2017; Shen et al., 2021; Pang et al., 2025). While general-purpose parallel translation data is abundantly available, automatically collecting a sufficient amount of domain-specific parallel data is challenging, and such special-purpose translation tends to require custom-made parallel data due to its specialized environment, e.g., terminologies in the medical domain or textual styles for a specific company, demanding a specialist to construct or check the quality of the parallel data (Chu & Wang, 2018; Yeganova et al., 2021).

In this study, we leverage easily accessible pre-trained NMT models and propose to adapt a general domain NMT model to a specific domain by using only the monolingual lexical distribution obtained from monolingual domain data in the target language. Lexical distribution, or word frequency, is crucial in analyzing the differences of corpora (Kilgarriff, 1997; Rayson & Garside, 2000), and the simple statistics have been investigated in various fields, such as psycholinguistic (Garlock et al., 2001) and language learning (Laufer & Nation, 1995), to quantify language usages. Our challenge lies in whether we can adapt a model to a new domain using the naive statistics, i.e., monolingual lexical distribution, which are already employed in various language assessments and easily estimated from the specific domain in the target language without consulting a specialist to translate or check the bilingual data qualities.

However, naively performing fine-tuning of a pre-trained NMT model and forcibly changing its token prediction distribution can lead to catastrophic forgetting issues, ranging from the loss of fluency (Korbak et al., 2022; Choshen et al., 2020; Kiegeland & Kreutzer, 2021) to degradation in non-specific domains (Saunders & DeNeefe, 2024; Gu & Feng, 2020; Thompson et al., 2019), thereby causing a reduction in translation performance. To achieve the domain shift while reducing catastrophic forgetting, i.e., harmlessly modifying the model's knowledge without degradation in generalization performance or excessive overfit to a specific domain, we represent the lexical distribution of the target domain as conditional energy-based models (EBMs) and approximate the EBMs using Conditional Distributional Policy Gradients (CDPG) (Korbak et al., 2022), which is a variant of the Generation under Distributional Control (GDC) framework (Khalifa et al., 2021). Korbak et al. (2022) had only verified the effectiveness of CDPG for small shifts, such as translating numeral nouns (e.g., "two") as digits (e.g., "2"), while our method employs sparse features, i.e., the monolingual lexical distribution of the target domain by treating each token-level statistic as a feature, enabling domain shifts in a fine-grained manner without catastrophic forgetting.

We confirm its effectiveness across several domain adaptation benchmarks (Tian et al., 2014; Koehn & Knowles, 2017; Aharoni & Goldberg, 2020) and scenarios, covering four language directions and four domain datasets, resulting in a total of 16 settings, thus we achieved unsupervised domain adaptation using only target-side domain data. Moreover, we propose DYNAMIC CDPG, which dynamically adjusts parameters using a small amount of bilingual validation data (or back-translated data in a fully unsupervised setting) to search for better configurations and analyze ideal settings for unsupervised domain adaptation. Our analysis of CDPG and DYNAMIC CDPG reveals that while selecting parameters sensitively can sometimes yield the best results, a simple CDPG can sufficiently achieve domain shift while reducing catastrophic forgetting.

To summarize this work, our contributions are as follows:

- We first demonstrate that unsupervised domain adaptation of neural machine translation can be performed robustly under a challenging yet realistic scenario where only a small amount of monolingual in-domain data is available. We leverage the lexical distribution, i.e., the word-frequency statistics of the target-domain data, and train a large number of EBMs within the CDPG framework to achieve robust domain shift without catastrophic forgetting and to demonstrate strong resilience to noise.

- We explore the applicability of CDPG to more realistic downstream tasks, i.e., domain translation. While prior GDC work has only been validated under small, pointwise shifts, we extend the framework to utilize large numbers of EBMs constructed from rich feature sets such as lexical distributions, demonstrating that large-scale domain shifts can be effectively modeled by CDPG.

- We propose DYNAMIC CDPG, which dynamically adjusts its configurations to explore better CDPG settings for unsupervised domain adaptation. This adaptive mechanism enables the model to better capture and align with the target-domain distribution. As a result, DYNAMIC CDPG can yield further improvements in some cases and achieve more robust domain shifts.

- We conduct a comprehensive evaluation across 16 experimental scenarios and thorough analyses, including lexical-distribution similarity, qualitative case studies, robustness to noise and randomness, and parameter investigations, to clarify when CDPG is the most effective. This provides a practical contribution for low-resource or sensitive domains where parallel data is scarce.

## 2 Conditional Distributional Policy Gradients

Conditional Distributional Policy Gradients (CDPG) (Korbak et al., 2022) approximates the generative probabilities of a language model to a target distribution while preventing catastrophic forgetting. It softly modifies the pre-trained parameters $\theta$ by slightly shifting its generation distribution using energy-based models (EBMs) through fine-tuning. We briefly summarize the minimal background needed to follow our method and defer further details to Korbak et al. (2022).

We define the pre-trained conditional language model $a(\boldsymbol{x}|\boldsymbol{c})$ where $\boldsymbol{c}$ is a context, i.e., an input source language sentence, and $\boldsymbol{x}$ is a sentence, i.e., in a target language, sampled from the entire distribution $\mathcal{X}$ given $\boldsymbol{c}$. We introduce an EBM $p_{\boldsymbol{c}}(\boldsymbol{x})$ as a controlled distribution defined as:

$$p_{\boldsymbol{c}}(\boldsymbol{x}) = \frac{1}{Z_{\boldsymbol{c}}} a(\boldsymbol{x}|\boldsymbol{c}) b(\boldsymbol{x}, \boldsymbol{c}), \tag{1}$$

where $Z_{\boldsymbol{c}} = \sum_{\boldsymbol{x} \in \mathcal{X}} a(\boldsymbol{x}|\boldsymbol{c}) b(\boldsymbol{x}, \boldsymbol{c})$ is a partition function that normalizes the entire EBM $p_{\boldsymbol{c}}(\boldsymbol{x})$, and $b(\boldsymbol{x}, \boldsymbol{c})$ is a control condition function which is 1 when a certain constraint is met, e.g., digits are used in $\boldsymbol{x}$ when translating numeral nouns in $\boldsymbol{c}$. When $b(\boldsymbol{x}, \boldsymbol{c})$ is reduced to a product of binary scorers $\phi_i(\boldsymbol{x}) \in \{0, 1\}$ in order to encode rich constraints as proposed by Khalifa et al. (2021), the EBM is formulated as:

$$p_{\boldsymbol{c}}^{point}(\boldsymbol{x}) = \frac{1}{Z_{\boldsymbol{c}}^{point}} a(\boldsymbol{x}|\boldsymbol{c}) \prod_i \phi_i(\boldsymbol{x}), \tag{2}$$

where "point" ($p_{\boldsymbol{c}}^{point}$ and $Z_{\boldsymbol{c}}^{point}$) indicates pointwise (binary) constraints.

However, such binary constraints can represent only two outcomes, namely whether a sentence satisfies the condition, or not. This limitation makes it impossible to express preferences over partial fulfillment, e.g., enforcing that a constraint should hold with probability 0.5. For instance, when attempting to mitigate gender bias in generated text, one may want the model to produce female and male entities with equal probability, that is, 0.5 for each. In narrative generation or stylistic control, one may further require skewed preferences such as generating certain attributes with probabilities 0.8 and 0.2. Such requirements cannot be encoded using binary scorers alone, because they only allow all-or-nothing fulfillment of the constraint.

To overcome this limitation, Khalifa et al. (2021) proposed a distributional constraint method for the unconditional case $p(\boldsymbol{x}) = \frac{1}{Z} a(\boldsymbol{x}) b(\boldsymbol{x})$, and Kruszewski et al. (2023) extended this idea to the conditional EBM with an exponential family form as follows:

$$p_{\boldsymbol{c}}^{dist}(\boldsymbol{x}|\boldsymbol{\lambda}) = \frac{1}{Z_{\boldsymbol{c}}^{dist}} a(\boldsymbol{x}|\boldsymbol{c}) \exp(\boldsymbol{\lambda} \cdot \boldsymbol{\phi}(\boldsymbol{x}, \boldsymbol{c})), \tag{3}$$

where $\boldsymbol{\lambda}$ is a parameter vector controlling the desired expected feature values encoded in $\boldsymbol{\phi}$. "dist" simply indicates the distributional version of the EBM, in contrast to the pointwise binary case above.

The parameter $\boldsymbol{\lambda}$ is determined through fine-tuning by starting from random initialization and iteratively updating by stochastic gradient descent (SGD) to minimize the loss function considering a distribution over contexts $\tau(\boldsymbol{c})$ as follows:

$$\nabla_{\boldsymbol{\lambda}} \mathcal{L}_{coef}(\boldsymbol{\lambda}) = \mathbb{E}_{\boldsymbol{c} \sim \tau(\boldsymbol{c})} \mathbb{E}_{\boldsymbol{x} \sim p_{\boldsymbol{c}}^{dist}(\boldsymbol{x}|\boldsymbol{\lambda})} \boldsymbol{\phi}(\boldsymbol{x}, \boldsymbol{c}) - \bar{\boldsymbol{\mu}}, \tag{4}$$

where $\bar{\boldsymbol{\mu}}$ is the probability vector specifying the desired feature distribution, and the inner expectation $\mathbb{E}_{\boldsymbol{x} \sim p_{\boldsymbol{c}}^{dist}(\cdot;\boldsymbol{\lambda})}$, i.e., the feature moments under the controlled EBM, is estimated through self-normalized importance sampling using $a(\cdot)$, following Korbak et al. (2022). In the previous example, if the neutral gender is expected, the desired probability is 0.5 for a female binary scorer. $\mathcal{L}_{coef}$ refers to the coefficient update objective for adjusting the parameter vector $\boldsymbol{\lambda}$ so that the controlled distribution matches the desired feature expectations.

However, since the EBM $p_{\boldsymbol{c}}(\boldsymbol{x})$ in Equation 1 and its exponential variant of Equation 3 is not an autoregressive language model, it cannot be used directly for inference. Therefore, training is conducted using the autoregressive model $\pi_{\theta}(\boldsymbol{x}|\boldsymbol{c})$ to approximate $p$ across contexts on average by minimizing the expected cross-entropy loss CE($\cdot$) between $\pi_{\theta}(\boldsymbol{x}|\boldsymbol{c})$ and multiple $p_{\boldsymbol{c}}$ of the EBM as follows:

$$\mathcal{L}(\theta) = \mathbb{E}_{\boldsymbol{c} \sim \tau(\boldsymbol{c})} \text{CE}\left(p_{\boldsymbol{c}}^{dist}(\cdot|\boldsymbol{\lambda}), \pi_{\theta}(\cdot \mid \boldsymbol{c})\right). \tag{5}$$

The gradient of this objective takes the following importance-weighted form:

$$\nabla_\theta \mathcal{L}(\theta) = \mathbb{E}_{\boldsymbol{c} \sim \tau(\boldsymbol{c})} \nabla_\theta \, \mathrm{CE}\left(p_{\boldsymbol{c}}^{dist}(\cdot | \boldsymbol{\lambda}), \pi_\theta(\cdot \mid \boldsymbol{c})\right) \tag{6}$$

$$= -\mathbb{E}_{\boldsymbol{c} \sim \tau(\boldsymbol{c})} \mathbb{E}_{\boldsymbol{x} \sim p_{\boldsymbol{c}}^{dist}(\boldsymbol{x}|\boldsymbol{\lambda})} \nabla_\theta \log \pi_\theta(\boldsymbol{x} \mid \boldsymbol{c}) \tag{7}$$

$$= -\mathbb{E}_{\boldsymbol{c} \sim \tau(\boldsymbol{c})} \mathbb{E}_{\boldsymbol{x} \sim \pi_\theta(\boldsymbol{x}|\boldsymbol{c})} \frac{p_{\boldsymbol{c}}^{dist}(\boldsymbol{x}|\boldsymbol{\lambda})}{\pi_\theta(\boldsymbol{x} \mid \boldsymbol{c})} \nabla_\theta \log \pi_\theta(\boldsymbol{x} \mid \boldsymbol{c}). \tag{8}$$

The loss is estimated via importance sampling from $\pi_\theta$. By iteratively applying these updates to $\theta$, $\pi_\theta$ can approximate the generative probability of the target EBM, enabling autoregressive generation. Note that CDPG is a fine-tuning method; thus, it does not introduce any changes to parameter size, model architecture, or inference speed. For clarity, we note that CDPG alternates between two steps: (i) updating $\boldsymbol{\lambda}$ so that the controlled EBM matches the desired feature expectations, and (ii) updating $\theta$ so that the autoregressive model approximates this controlled distribution. These two coupled updates constitute the core mechanism, and full derivations can be found in Korbak et al. (2022).

## 3 Domain Adaptation by CDPG

### 3.1 Adaptation to Monolingual Lexical Distribution

We leverage monolingual data in the specific domain in the target language, e.g., English reports in the medical domain, and propose domain adaptation for NMT with CDPG using only the subword frequency information as features. When applying CDPG for NMT, the source sentence corresponds to a context $\boldsymbol{c}$ drawn from the source language in general domain, e.g., German news, and the ideal target sentence is derived from $p_{\boldsymbol{c}}^{dist}(\boldsymbol{x}|\boldsymbol{\lambda})$. For training CDPG under distribution constraints in Equation 3, it requires a binary scorer $\phi_i(\boldsymbol{x}, \boldsymbol{c})$ and a parameter $\lambda_i$ for each feature.

Under this setting, we employ $\boldsymbol{\phi}(\boldsymbol{x}, \boldsymbol{c})$ as features whether each subword of the target domain is included in $\boldsymbol{x}$. Moreover, when learning the parameter vector $\boldsymbol{\lambda}$ in Equation 4, we set the probability of each constraint, $\bar{\boldsymbol{\mu}}$, as the ratio of the frequency of subwords in the whole text in the target domain as follows:

$$\bar{\mu}_i = \frac{Freq^{target}(x_i)}{\sum_{x_j \in \mathbb{X}} Freq^{target}(x_j)}, \tag{9}$$

where $Freq^{target}$ denotes the frequency of each subword $x_i$ in the target text in the vocabulary $\mathbb{X}$. By performing the above operations, we attempt to address the domain shift by utilizing the frequency of all subwords of the target domain text. Since this feature selection only uses data from the target side, the creation of the EBM model only requires the target side domain text.

Note that the computational cost of each CDPG iteration is dominated by two components: (i) estimating the feature moments $\mathbb{E}_{\boldsymbol{x} \sim p_{\boldsymbol{c}}^{dist}(\cdot;\boldsymbol{\lambda})} \boldsymbol{\phi}(\boldsymbol{x}, \boldsymbol{c})$ in Equation 4, and (ii) the autoregressive LM update in Equation 5. For $N$ contexts and $M$ samples per context, the moment estimator evaluates $\boldsymbol{\phi}(\boldsymbol{x}, \boldsymbol{c})$ only for the subwords that actually appear in each sampled sequence $\boldsymbol{x}$. Consequently, the per-sample cost scales with the sequence length $L$, rather than with the full feature dimension $|\boldsymbol{\phi}|$. Intuitively, constructing the sparse lexical histogram for a sampled sequence is $O(L)$, and computing the importance weights over the $M$ samples in a context is linear in $M$. Therefore, the overall EBM-side complexity across $N$ contexts is $O(NML)$. Moreover, because the feature scorers factor across tokens and samples, the EBM-side computations parallelize trivially over $(\boldsymbol{c}, \boldsymbol{x})$. The LM update, i.e., the importance-weighted forward/backward passes of $\pi_\theta$, is also $O(NML)$ and typically dominates the wall-clock time. Thus, despite the large ambient feature space, the effective per-iteration complexity remains $O(NML)$. We operate on sparse, sequence-derived features that admit efficient vectorized and parallel computation, avoiding any need to process the full vocabulary-sized feature vector.

### 3.2 Dynamic CDPG

EBM is iteratively updated by Equation 4 to approximate the generative language model toward the expected probability distribution of the target domain. The procedure involves generating multiple sentences

---

**Algorithm 1:** Dynamic CDPG (Sketch)

---

**Given** : Dev set $\mathcal{D}_{\text{dev}}$; target features $\bar{\boldsymbol{\mu}}$; feature extractor $\boldsymbol{\phi}$; metric $\mathcal{M}$; outer iters $T$; LRs $\eta_\lambda, \eta_\theta$;
schedules $\mathbb{A} = [0.5, 0.4, 0.3, 0.2, 0.1]$, $\mathbb{B} = [0.6, 0.7, 0.8, 0.9, 1.0]$

**Input** : Pre-trained LM $a(\boldsymbol{x} \mid \boldsymbol{c})$; Conditional EBM $p_{\boldsymbol{c}}^{\text{dist}}(\boldsymbol{x} \mid \boldsymbol{\lambda})$

**Output** : Fine-tuned autoregressive model $\pi_\theta$

---

1   $\pi_\theta \leftarrow a$;   initialize $\boldsymbol{\lambda}$;   $S \leftarrow \mathbb{A}$;   $i \leftarrow 1$;   $p \leftarrow S[i]$;   $s^\star \leftarrow -\infty$          `// initialization`

2   **for** $t \leftarrow 1$ **to** $T$ **do**

     `// (1) EBM coefficients update (Equation 4)`

3     Sample contexts $\{\boldsymbol{c}\} \sim \tau(\boldsymbol{c})$;

4     Estimate $\mathbb{E}_{\boldsymbol{x} \sim p_{\boldsymbol{c}}^{\text{dist}}}[\boldsymbol{\phi}]$ via self-normalized IS using nucleus sampling $\boldsymbol{x} \sim a(\cdot \mid \boldsymbol{c};\ \text{top-}p = p)$;

5     $\boldsymbol{\lambda} \leftarrow \boldsymbol{\lambda} - \eta_\lambda\big(\mathbb{E}[\boldsymbol{\phi}] - \bar{\boldsymbol{\mu}}\big)$          `// push to target features (Equation 4)`

     `// (2) Autoregressive model update (Equation 5)`

6     Sample $(\boldsymbol{c}, \boldsymbol{x})$ with $\boldsymbol{x} \sim \pi_\theta(\cdot \mid \boldsymbol{c})$;

7     $w \leftarrow \dfrac{p_{\boldsymbol{c}}^{\text{dist}}(\boldsymbol{x})}{\pi_\theta(\boldsymbol{x} \mid \boldsymbol{c})}$;

8     $\theta \leftarrow \theta - \eta_\theta\, w\, \nabla_\theta \log \pi_\theta(\boldsymbol{x} \mid \boldsymbol{c})$          `// importance-weighted update (Equation 5)`

     `// (3) Dev evaluation and schedule update (Dynamic core part)`

9     $s \leftarrow \mathcal{M}(\pi_\theta, \mathcal{D}_{\text{dev}})$          `// e.g., BLEU`

10    **if** $s \geq s^\star$ **then**

11      $s^\star \leftarrow s$; $i \leftarrow i + 1$          `// accept: advance within S`

12    **else**

13      $S \leftarrow \begin{cases} \mathbb{B}, & S = \mathbb{A} \\ \mathbb{A}, & S = \mathbb{B} \end{cases}$; $i \leftarrow 1$          `// reject: switch schedule`

14    **if** $i > |S|$ **then**

15      **break**          `// stop when a schedule is exhausted`

16    $p \leftarrow S[i]$          `// set next top-p`

17 **return** $\pi_\theta$

---

$\boldsymbol{x}$ conditioned on context $\boldsymbol{c}$. To sample $\boldsymbol{x}$ from the entire distribution $\mathcal{X}$ given $\boldsymbol{c}$, we often employ nucleus sampling or ancestral sampling as a decoding strategy (Khalifa et al., 2021; Meng et al., 2022; Korbak et al., 2022). Nucleus sampling (Holtzman et al., 2020) selects tokens in descending order of probability until their cumulative probability mass reaches a threshold $p$, and then samples from this set according to the multinomial distribution. The nucleus sampling parameter top-$p$ controls the diversity of the generated outputs, and when $p = 1$, i.e., ancestral sampling, it produces fully diverse samples from the full multinomial distribution. A higher top-$p$ produces more diverse outputs, whereas a lower top-$p$ yields more conservative ones.

However, when there is a large gap between the distribution of the pre-trained model and that of the target domain, simple ancestral sampling, which produces fully diverse samples, is not effective (Eikema et al., 2022; Kim et al., 2025). In practice, modifying the sampling distribution by changing the sampling method or adjusting its parameters can make sampling more effective in practical settings (Go et al., 2023; Freitag et al., 2023; Nadeem et al., 2020; Kamigaito et al., 2025). Therefore, the sampling diversity must be controlled to obtain better contrastive samples for precise scoring, and CDPG should apply different top-$p$ settings for different domains, e.g., higher top-$p$ for larger gap in distance, and lower top-$p$ values for closer domains. Furthermore, when the EBM becomes closer to the target domain as training proceeds, a lower value of top-$p$ will be sufficient, but it still demands a higher value of top-$p$ for the larger distance. Therefore, we introduce DYNAMIC CDPG, which dynamically adjusts top-$p$ at each iteration of the approximation to the EBM in Equation 1, searching for better configurations of CDPG.

Algorithm 1 outlines the procedure of DYNAMIC CDPG. DYNAMIC CDPG leverages a bilingual development set[1] to monitor training progress and guide adjustments to the sampling strategy. The core idea of DYNAMIC

---

[1]The development set refers to the text used to generate features. These data can be constructed from the target-side monolingual corpus via back-translation, as described later. Therefore, the development set remains completely disjoint from the test data, ensuring that it functions as a proper held-out validation set.

CDPG is to divide the training process into several iterations, then start with a constant parameter for top-$p$, and reconsider it in each training iteration such that a smaller top-$p$ will be selected in the next iteration if a larger top-$p$ leads to inferior performance on the development set (lines 10-17). Note that if synthetic bilingual data can be obtained via back-translation from the target in-domain data, the setting can still be regarded as fully unsupervised. For each iteration, we use an evaluation method, such as BLEU (Papineni et al., 2002), to assess the model's performance (line 10) and decide whether to accept the update.

Specifically, we heuristically define two candidate schedules for top-$p$: $\mathbb{A} = [0.5, 0.4, 0.3, 0.2, 0.1]$ (descending), which gradually reduces sampling diversity and encourages the model to fit more closely to the target features, and $\mathbb{B} = [0.6, 0.7, 0.8, 0.9, 1.0]$ (ascending), which increases sampling diversity and serves as a fallback exploration schedule. We start the iteration with the first element of $\mathbb{A}$ as the value of top-$p$. If an iteration is accepted, the algorithm proceeds to the next iteration with the second element of $\mathbb{A}$ (line 12); if rejected, the algorithm switches to $\mathbb{B}$ (line 14) and continues through its elements. The process terminates once all values in either $\mathbb{A}$ or $\mathbb{B}$ have been exhausted (lines 15-16). Our preliminary studies showed that the training under DYNAMIC CDPG is always stable under our top-$p$ scheduling.

## 4 Experimental Setup

### 4.1 Datasets

We conduct experiments with four translation pairs of English to German (en→de), German to English (de→en), English to Chinese (en→zh), and Chinese to English (zh→en). For pairs involving de, we collect four domains, including IT, Medical, Law, and Koran from the public corpus[2] released by Koehn & Knowles (2017); Aharoni & Goldberg (2020), where each domain has 2,000 sentences for the development set and test set, respectively[3]. For pairs involving zh, we collect four domains, including Education, Laws, Thesis, and Science, from the UM-Corpus (Tian et al., 2014), which is public[4] with high quality. Although this corpus provides $456 - 790$ sentences for test sets in those 4 domains, the development set is not provided. Therefore, we randomly select 3,000 sentences from the training data for each domain as the development sets. Moreover, we use the development sets[5] of WMT from $2018 - 2022$ (Bojar et al., 2018; Barrault et al., 2019; 2020; Akhbardeh et al., 2021; Kocmi et al., 2022), i.e., 14,482 translation instances of the newsdev set from a news domain, to train CDPG for all translation directions by treating them as a generic domain data set. Specifically, the contexts $\tau(\boldsymbol{c})$ are collected from the 14,482 source language sentences of the newsdev set, and the domain features $\bar{\boldsymbol{\mu}}$ are derived from the target language sentences of the domain-specific instances.

### 4.2 Models

**Baselines**  We employ four open-source NMT models (Tiedemann & Thottingal, 2020) from HuggingFace Transformers[6] (Wolf et al., 2020) as backbones in our experiments, denoted as PRE-TRAINED. These models are based on the Transformer architecture (Vaswani et al., 2017) and are trained on OPUS (Tiedemann, 2012) with the same configuration[7] comprising the 6 encoder and decoder layers, 8 attention heads, embedding size of 512, and an inner size of 2048. Since CDPG fine-tunes all parameters, we first establish a baseline by naively fine-tuning the model on the development sets, denoted as FINE-TUNED. We also adopt LoRA (Hu et al., 2022) as a second baseline, which adapts attention weights with an inner rank of 8. The baseline is then fine-tuned using the same setup as FINE-TUNED, offering robustness and reduced susceptibility to catastrophic forgetting when working with smaller datasets (Xu et al., 2023; Biderman et al., 2024). We set the batch size to 128 and the learning rate to 2e-7 for both settings with the Adam optimizer (Kingma & Ba, 2017). As another baseline, we conduct back-translation (Sennrich et al., 2016), denoted as BACK-TRANS.

---

[2] https://github.com/roeeaharoni/unsupervised-domain-clusters

[3] Given the low quality of this corpus, we clean it and re-align the test set using de as the basis to avoid potential bias in evaluation. The low quality includes but is not limited to repetition, no alignment, and noise. Furthermore, the refined test data is completely unseen, enabling evaluation without any data contamination issues in the existing training corpus (Raunak & Menezes, 2022). The cleaned datasets are publicly available at https://github.com/naist-nlp/cdpg.

[4] http://nlp2ct.cis.umac.mo/um-corpus/

[5] http://data.statmt.org/wmt23/general-task/dev.tgz

[6] https://huggingface.co/Helsinki-NLP

[7] Details in: https://huggingface.co/Helsinki-NLP/opus-mt-en-zh/blob/main/config.json

Specifically, we generate source-language data using a reverse-direction model and the same target-language data used for fine-tuning. The model is then fine-tuned on the generated data using the same settings. All fine-tuning experiments are conducted for 10 epochs. The checkpoint achieving the best performance on the development set is selected as baselines.

**CDPG**   For training the parameter vector $\boldsymbol{\lambda}$ in Equation 4, we set the batch size to 8 and the learning rate $\eta_\lambda$ to 0.05 with a constant learning rate scheduler based on the training loss in our preliminary studies. Likewise, for fine-tuning CDPG model parameters $\theta$ in Equation 5, we set the batch size to 128, the number of epochs to 10, and the learning rate $\eta_\theta$ to 2e-5 with a constant learning rate scheduler and the Adam optimizer. We always set top-$p$ to 0.5 in training $\boldsymbol{\lambda}$ and fine-tuning $\theta$. Moreover, we set the character length of the considered features, i.e., subwords, to be no less than 3 to filter insignificant features, and the input texts are pre-processed by the tokenizer in each pre-trained model. We used *disco* (Kruszewski et al., 2023) to implement the CDPG.

**Dynamic CDPG**   We maintain the hyperparameters of CDPG for DYNAMIC CDPG. DYNAMIC CDPG additionally uses a bilingual development set to guide the dynamic selection of the top-$p$ value at each iteration, as described in Section 3.2. We construct this development set via back-translation from the target-side monolingual corpus, similar to BACK-TRANS. Therefore, the procedure remains fully unsupervised and is completely isolated from the test data, just as in CDPG. In our preliminary experiment comparing back-translated bilingual data and the actual bilingual development set, the tuning results were the same. One possible reason is that the target-side lexical distribution plays a more important role than the source-side or full parallel data. Thus, we use back-translated data for DYNAMIC CDPG while maintaining the unsupervised setting. We set each iteration of DYNAMIC CDPG to 10 epochs. We use BLEU (Papineni et al., 2002) to calculate the validation score for each epoch. Additionally, we set a threshold that requires at least three improvements in the validation score for an iteration to be accepted. The initial learning rate of each subsequent iteration is set by dividing that of the previously accepted iteration by the square root of the number of epochs, to ensure training stability.

### 4.3   Evaluation

We set the beam size of 4 for each model to generate translations for the entire test set, and did not employ nucleus sampling (Holtzman et al., 2020) in the final evaluation for consistent evaluation in all settings. Then, translations are evaluated by four automatic MT evaluation methods: 1) Confidence (Müller et al., 2019; Wang et al., 2020), calculated as the average of the Softmax probabilities assigned to each generated token by the NMT system. 2) BLEU (Papineni et al., 2002), assessed with the implementation of SacreBLEU (Post, 2018) to measure the surface-level similarities, 3) NIST (Doddington, 2002), which is similar to BLEU but gives special attention to low-frequency words to assess the qualities of domain-specific terminologies, and 4) BERTScore (Zhang et al., 2020), which reports embedding similarities by Precision, Recall, and F1 scores, where the F1 score being the harmonic mean of Precision and Recall.

## 5   Experimental Results

Table 1 shows the experimental results for en→de and de→en translation pairs, and Table 2 shows the results for en→zh and zh→en translation pairs.

**Finding 1: Naive supervised fine-tuning methods (almost) fail at domain adaptation with limited in-domain data.**   First, compared to PRE-TRAINED, both FINE-TUNED and LoRA, which naively utilize supervision signals from parallel in-domain sentences, generally fail to achieve improvements, except for slight gains in *Medical* of en→de, *Laws* of en→zh, and *Thesis* and *Science* of zh→en. Furthermore, BACK-TRANS, which synthesizes pseudo-parallel data for fine-tuning, often performs comparably or worse than FINE-TUNED, which uses a small amount of curated in-domain data. These results indicate that in limited-resource settings, such as domain adaptation, naive adaptation from a pure NMT model, such as PRE-TRAINED, not only fails to improve performance but often leads to degradation.

Table 1: Our main evaluation results for the `en→de` and `de→en` translation pairs. Pre-trained indicates the performance of the original models without any fine-tuning, while the other methods are described in Section 4.2. Conf. is the abbreviation of Confidence, and P and R represent the Precision and Recall scores of BERTScore, respectively. The best score in each block, which is divided by domain and language pair, is highlighted in **bold**. The checkmark (✓) indicates that parallel in-domain sentences were used as supervision signals. Identical scores between Dynamic CDPG and CDPG, such as Koran domain in `de→en`, indicate that Dynamic CDPG rejected all updates after initialization. Red cells indicate improvements from the base model Pre-trained to the CDPG-based model.

| Domain | | Method | en→de | | | | | | de→en | | | | | |
|---|---|---|---|---|---|---|---|---|---|---|---|---|---|---|
| | | | Conf. | BLEU | NIST | P | R | F1 | Conf. | BLEU | NIST | P | R | F1 |
| IT | | Pre-trained | 68.39 | 27.58 | 5.97 | 87.48 | 87.70 | 87.52 | 72.02 | 38.80 | 7.96 | 94.93 | 94.92 | 94.91 |
| | ✓ | Fine-tuned | 67.91 | 27.92 | 6.04 | 87.38 | 87.60 | 87.42 | 71.76 | 38.83 | 7.95 | 94.94 | 94.93 | 94.92 |
| | ✓ | LoRA | 67.79 | 26.88 | 5.83 | 87.33 | 87.56 | 87.37 | 71.46 | 38.32 | 7.86 | 94.92 | 94.91 | 94.91 |
| | | Back-trans | 67.90 | 27.89 | 6.01 | 87.38 | 87.59 | 87.41 | 71.49 | 38.35 | 7.86 | 94.93 | 94.92 | 94.91 |
| | | CDPG | 74.44 | 29.01 | 6.25 | 87.68 | 87.77 | 87.67 | **77.91** | 39.79 | 8.30 | 94.95 | 94.94 | 94.93 |
| | | Dynamic CDPG | **79.36** | **30.78** | **6.58** | **88.00** | **87.87** | **87.89** | 77.65 | **40.55** | **8.34** | **95.01** | **94.96** | **94.98** |
| Medical | | Pre-trained | 75.93 | 43.19 | 8.45 | 91.55 | 91.17 | 91.31 | 78.06 | **45.50** | 8.47 | **96.65** | 96.50 | **96.57** |
| | ✓ | Fine-tuned | 75.71 | 43.23 | 8.46 | 91.53 | 91.14 | 91.29 | 77.77 | 45.48 | 8.47 | 96.64 | **96.50** | 96.56 |
| | ✓ | LoRA | 75.50 | **43.56** | 8.52 | 91.55 | 91.15 | 91.30 | 77.72 | 44.31 | 8.35 | 96.61 | 96.49 | 96.54 |
| | | Back-trans | 75.50 | 43.56 | 8.52 | 91.57 | 91.17 | 91.32 | 77.65 | 45.47 | 8.55 | 96.64 | 96.49 | 96.56 |
| | | CDPG | 80.85 | 42.54 | **8.60** | **91.61** | **91.28** | **91.40** | **82.84** | 44.56 | **8.56** | 96.57 | **96.50** | 96.53 |
| | | Dynamic CDPG | **82.32** | 43.51 | 8.54 | 91.60 | 91.20 | 91.36 | **82.84** | 44.56 | **8.56** | 96.57 | **96.50** | 96.53 |
| Law | | Pre-trained | 72.49 | 44.82 | 9.01 | 89.38 | 89.11 | 89.22 | 72.89 | **51.75** | 10.05 | 96.06 | **95.75** | **95.90** |
| | ✓ | Fine-tuned | 72.08 | 44.83 | 9.01 | 89.39 | 89.10 | 89.22 | 72.53 | 51.70 | 10.04 | 96.06 | 95.74 | 95.89 |
| | ✓ | LoRA | 72.05 | 44.80 | 9.01 | **89.42** | 89.12 | **89.25** | 72.55 | 51.67 | 10.04 | 96.05 | 95.73 | 95.89 |
| | | Back-trans | 72.05 | 44.62 | 8.97 | 89.39 | 89.10 | 89.22 | 72.45 | 51.69 | 10.05 | 96.06 | 95.74 | 95.89 |
| | | CDPG | 77.36 | 44.12 | **9.05** | 89.33 | **89.17** | 89.22 | **78.12** | 51.61 | 10.12 | 96.02 | 95.72 | 95.86 |
| | | Dynamic CDPG | **78.18** | **44.87** | 9.03 | 89.40 | 89.09 | 89.22 | 73.02 | 51.64 | **10.15** | **96.07** | 95.73 | 95.89 |
| Koran | | Pre-trained | 61.51 | **18.90** | 5.25 | 81.59 | **80.18** | 80.84 | 59.23 | 20.86 | 5.66 | **91.95** | 91.07 | **91.49** |
| | ✓ | Fine-tuned | 61.39 | 18.86 | 5.24 | 81.56 | 80.16 | 80.82 | 58.80 | 20.81 | 5.65 | 91.94 | 91.06 | 91.48 |
| | ✓ | LoRA | 61.18 | 18.86 | 5.24 | 81.54 | 80.13 | 80.80 | 58.94 | 20.83 | 5.65 | 91.94 | 91.05 | 91.48 |
| | | Back-trans | 61.15 | 18.84 | 5.24 | 81.53 | 80.13 | 80.79 | 58.78 | 20.79 | 5.65 | 91.94 | 91.06 | 91.49 |
| | | CDPG | **67.00** | 18.40 | **5.26** | 81.46 | 80.06 | 80.72 | **64.75** | 20.94 | **5.67** | 91.90 | **91.09** | 91.48 |
| | | Dynamic CDPG | 61.30 | 18.85 | 5.25 | **81.63** | 80.16 | **80.85** | **64.75** | **20.94** | **5.67** | 91.90 | **91.09** | 91.48 |

**Finding 2: CDPG improves confidences and performances on low-frequency, domain-specific terms.** Next, compared to Pre-trained, although CDPG consistently improves Confidence, its performance in BLEU and BERTScore varies across domains. Specifically, in domains such as *IT* of `en↔de` and *Education* of `en↔zh`, where other baseline methods show only limited gains, CDPG achieves improvements across all metrics. In contrast, in domains like *Law* and *Koran* of `en↔de`, *Medical* of `de→en`, and *Laws* of `en→zh`, the improvements in BLEU and BERTScore are limited. However, NIST scores, which place greater emphasis on low-frequency words, still improve in all cases. This indicates that the consistent increases in Confidence and NIST scores are due to improved handling of domain-specific terms, which are often struggle to capture in general MT evaluation metrics such as BLEU because of their relatively low frequency. Therefore, compared to the baselines, domain adaptation using CDPG is shown to be more robust, even in the settings where access to parallel in-domain data is limited or unavailable.

**Finding 3: While Dynamic CDPG often provides slight improvements, CDPG with fixed parameters already demonstrates sufficient performance.** Dynamic CDPG aims to improve MT metrics, such as BLEU, by dynamically controlling and selecting parameters based on evaluations using a small amount of parallel in-domain development data at each epoch. Since BLEU is used as the guiding signal, Dynamic CDPG consistently achieves higher BLEU scores than CDPG. In domains such as *IT* of `en↔de`, *Thesis* of `en→zh`, and *Education* of `zh→en`, Dynamic CDPG selects parameters that further

Table 2: Our main evaluation results for the `en→zh` and `zh→en` translation pairs. Notation and other corresponding information are the same as in Table 1.

| Domain | | Method | en→zh | | | | | | zh→en | | | | | |
|---|---|---|---|---|---|---|---|---|---|---|---|---|---|---|
| | | | Conf. | BLEU | NIST | P | R | F1 | Conf. | BLEU | NIST | P | R | F1 |
| Education | | Pre-trained | 49.88 | 30.26 | 0.73 | 83.82 | 82.18 | 82.94 | 60.15 | 23.49 | 5.56 | 94.44 | 94.16 | 94.30 |
| | ✓ | Fine-tuned | 49.28 | 30.07 | 0.68 | 83.70 | 81.96 | 82.78 | 59.63 | 23.54 | 5.56 | 94.43 | 94.16 | 94.29 |
| | ✓ | LoRA | 49.03 | 30.19 | 0.68 | 83.70 | 81.92 | 82.75 | 59.64 | 23.69 | 5.57 | 94.49 | 94.16 | 94.30 |
| | | Back-trans | 49.13 | 30.04 | 0.67 | 83.69 | 81.91 | 82.74 | 59.65 | 23.48 | 5.56 | 94.44 | 94.15 | 94.29 |
| | | CDPG | **57.88** | 31.03 | 0.93 | 84.59 | **83.23** | **83.86** | 66.05 | 23.69 | 5.60 | 94.52 | **94.28** | 94.40 |
| | | Dynamic CDPG | 57.22 | **31.16** | **0.94** | 84.71 | 83.01 | 83.81 | **67.02** | 24.23 | **5.67** | 94.60 | 94.28 | **94.44** |
| Laws | | Pre-trained | 62.06 | 51.73 | 0.59 | 89.67 | **89.70** | 89.65 | 63.84 | 32.36 | 6.11 | 94.55 | 93.52 | 94.02 |
| | ✓ | Fine-tuned | 61.46 | 51.71 | 0.59 | 89.74 | **89.70** | **89.69** | 63.47 | 32.27 | 6.10 | 94.52 | 93.49 | 93.99 |
| | ✓ | LoRA | 61.38 | **51.87** | 0.60 | **89.75** | 89.63 | 89.66 | 63.16 | 32.33 | 6.09 | 94.51 | 93.45 | 93.97 |
| | | Back-trans | 61.26 | 51.61 | 0.56 | 89.75 | 89.69 | 89.69 | 63.18 | 32.22 | 6.07 | 94.52 | 93.46 | 93.97 |
| | | CDPG | **68.50** | 50.81 | **0.68** | 89.60 | 89.65 | 89.60 | 69.68 | 34.54 | 6.34 | 94.68 | 93.77 | 94.21 |
| | | Dynamic CDPG | **68.50** | 50.81 | **0.68** | 89.60 | 89.65 | 89.60 | **70.50** | **35.06** | **6.38** | **94.74** | **93.87** | **94.29** |
| Thesis | | Pre-trained | 47.62 | 18.95 | 1.14 | 76.09 | 75.69 | 75.78 | 50.83 | 8.65 | 3.48 | 89.55 | 88.33 | 88.92 |
| | ✓ | Fine-tuned | 47.23 | 19.94 | 1.39 | 76.42 | **75.75** | 75.99 | 50.11 | 8.60 | 3.46 | 89.56 | 88.31 | 88.91 |
| | ✓ | LoRA | 47.22 | 19.34 | 1.25 | 76.36 | 75.72 | 75.93 | 50.15 | **8.71** | 3.48 | 89.58 | 88.33 | 88.93 |
| | | Back-trans | 46.98 | 19.95 | 1.36 | 76.40 | 75.70 | 75.95 | 50.05 | 8.67 | 3.47 | 89.58 | 88.30 | 88.92 |
| | | CDPG | **54.19** | 19.94 | 1.29 | 76.11 | 75.53 | 75.72 | 57.16 | 8.49 | 3.51 | 89.52 | 88.38 | 88.93 |
| | | Dynamic CDPG | 51.22 | **20.14** | **1.52** | **76.53** | 75.72 | **76.03** | **58.57** | 8.53 | **3.54** | **89.67** | 88.37 | **89.00** |
| Science | | Pre-trained | 47.56 | 24.45 | 0.94 | 81.28 | 79.06 | 80.09 | 57.97 | 16.20 | 4.86 | 92.80 | 92.60 | 92.69 |
| | ✓ | Fine-tuned | 47.00 | 24.52 | 0.94 | 81.26 | 79.05 | 80.07 | 57.48 | **16.36** | **4.88** | **92.82** | 92.60 | **92.70** |
| | ✓ | LoRA | 46.75 | 24.57 | 0.96 | 81.38 | 79.09 | 80.15 | 57.49 | 16.29 | **4.88** | 92.81 | 92.60 | **92.70** |
| | | Back-trans | 46.77 | 24.46 | 0.93 | 81.26 | 79.07 | 80.08 | 57.48 | 16.33 | 4.87 | 92.81 | 92.60 | **92.70** |
| | | CDPG | **56.27** | 24.78 | **1.02** | 81.48 | **79.70** | **80.53** | 64.06 | 15.96 | **4.88** | 92.76 | **92.66** | **92.70** |
| | | Dynamic CDPG | 52.38 | **24.80** | 1.00 | **81.63** | 79.39 | 80.43 | **65.55** | 16.34 | 4.86 | 92.79 | 92.60 | 92.69 |

Table 3: The top-$p$ values used in Dynamic CDPG. Values are presented in the order they are used.

| | IT | Medical | Law | Koran |
|---|---|---|---|---|
| en→de | 0.5,0.4,0.8 | 0.5,0.7,1.0 | 0.5,0.8 | 1.0 |
| de→en | 0.5,0.9 | 0.5 | 0.5,0.9 | 0.5 |

| | Education | Laws | Thesis | Science |
|---|---|---|---|---|
| en→zh | 0.5,0.9 | 0.5 | 0.5,0.7 | 0.5,0.6,0.7 |
| zh→en | 0.5,0.4 | 0.5,0.4 | 0.5,0.6,0.7,0.8 | 0.5,0.3,0.2,0.1 |

boost the improvements already observed in CDPG. Moreover, Dynamic CDPG mitigates the performance degradation seen with CDPG in some domains, including *Medical*, *Law*, and *Koran* of `en→de`, and *Science* of `zh→en`. Table 3 summarizes the top-$p$ values selected and how they changed during training. In *Medical* and *Koran* of `de→en` and *Laws* of `en→zh`, the selected top-$p$ remained identical to the default CDPG setting, resulting in the same scores reported in Tables 1 and 2. These findings suggest that while Dynamic CDPG can yield further improvements under metric supervision, the fixed-parameter CDPG already delivers strong and stable results. Therefore, although parameter tuning is ideal, CDPG is not overly sensitive to the top-$p$ value, and a fixed value of 0.5 is generally sufficient.

# 6 Discussion

## 6.1 When Is CDPG Effective?

Based on the results in Tables 1 and 2, and the discussion in Section 5, although CDPG-based methods consistently improved Confidence and NIST, performance fluctuations were observed depending on each domain. To better understand under what conditions CDPG is effective, we conduct a detailed analysis focusing on the distributional differences between the pre-trained models and the monolingual lexical distribution features of CDPG described in Section 3.1.

Table 4: Comparisons of the target-side lexical distribution features. The prefix *len.* indicates the number of features, and *sim.* denotes the cosine similarity (in percent). *len.1* and *len.2* refer to the lengths of the first and second feature sets in each comparison. *itr* and *uni* indicate that the similarity was computed over the intersection or the union of the two sets, respectively.

| Pair | Domain | Case (i): *Dev Features* v.s. *Pretrained Features* | | | | | | Case (ii): *Dev Features* v.s. *Test Features* | | | | | |
|---|---|---|---|---|---|---|---|---|---|---|---|---|---|
| | | len.1 | len.2 | len.itr | **sim.itr** | len.uni | **sim.uni** | len.1 | len.2 | len.itr | **sim.itr** | len.uni | **sim.uni** |
| **en→de** | IT | 5,832 | 5,553 | 4,152 | 83.14 | 7,233 | 65.09 | 5,832 | 5,475 | 3,366 | 93.14 | 7,941 | 90.99 |
| **en→de** | Koran | 4,543 | 3,948 | 2,931 | 95.69 | 5,560 | 95.48 | 4,543 | 4,435 | 3,300 | 98.81 | 5,678 | 98.67 |
| **de→en** | Law | 7,054 | 6,469 | 5,668 | 98.80 | 7,855 | 98.66 | 7,054 | 7,014 | 4,754 | 98.19 | 9,314 | 97.91 |
| **de→en** | Medical | 6,543 | 6,130 | 5,367 | 95.83 | 7,306 | 94.97 | 6,543 | 6,577 | 4,604 | 94.22 | 8,516 | 93.48 |
| **en→zh** | Thesis | 7,533 | 7,518 | 5,395 | 74.88 | 9,656 | 73.23 | 7,533 | 3,755 | 3,188 | 92.81 | 8,100 | 91.82 |
| **en→zh** | Laws | 6,903 | 6,783 | 4,865 | 68.11 | 8,821 | 64.99 | 6,903 | 1,852 | 1,373 | 29.43 | 7,382 | 24.64 |
| **zh→en** | Education | 10,680 | 9,546 | 7,379 | 80.64 | 12,847 | 79.92 | 10,680 | 2,357 | 1,885 | 70.37 | 10,711 | 65.13 |
| **zh→en** | Science | 9,807 | 9,127 | 6,866 | 61.38 | 12,068 | 60.64 | 9,807 | 3,089 | 2,317 | 65.39 | 11,692 | 56.79 |

Table 5: Results of CDPG with fixed top-*p* values of 0.5, 0.8, and 1.0. The results for top-*p* = 0.5 are the same as those reported in Tables 1 and 2, and the abbreviations are consistent with those tables. The best score in each row is highlighted in **bold**. The row order aligns with Table 4.

| Pair | Domain | top-*p* = 0.5 | | | | top-*p* = 0.8 | | | | top-*p* = 1.0 | | | |
|---|---|---|---|---|---|---|---|---|---|---|---|---|---|
| | | Conf. | BLEU | NIST | F1 | Conf. | BLEU | NIST | F1 | Conf. | BLEU | NIST | F1 |
| **en→de** | IT | 74.44 | 29.01 | 6.25 | **87.67** | **74.67** | **29.13** | **6.28** | 87.66 | 67.87 | 28.19 | 6.08 | 87.47 |
| **en→de** | Koran | 67.00 | 18.40 | 5.25 | 80.72 | **67.14** | 18.50 | 5.29 | 80.74 | 61.30 | **18.85** | **5.35** | **80.85** |
| **de→en** | Law | 78.12 | **51.61** | 10.12 | 95.86 | **78.33** | 51.53 | **10.16** | 95.86 | 71.83 | 51.58 | **10.16** | **95.87** |
| **de→en** | Medical | 82.84 | 44.56 | 8.56 | 96.53 | **83.06** | 44.82 | **8.61** | **96.54** | 77.72 | **45.06** | **8.61** | **96.54** |
| **en→zh** | Thesis | **54.19** | 19.94 | 1.29 | 75.72 | 53.93 | **19.98** | 1.48 | **75.86** | 46.96 | 19.95 | 1.47 | 75.76 |
| **en→zh** | Laws | 68.50 | 50.81 | **0.68** | 89.60 | **68.78** | 51.16 | 0.65 | 89.63 | 61.68 | **51.90** | 0.61 | **89.71** |
| **zh→en** | Education | **66.05** | 23.69 | 5.60 | **94.40** | 65.86 | **23.92** | **5.65** | 94.37 | 59.68 | 23.50 | 5.58 | 94.31 |
| **zh→en** | Science | **64.06** | 15.96 | **4.88** | **92.70** | 63.93 | 16.14 | 4.87 | **92.70** | 57.29 | **16.34** | **4.88** | 92.69 |

First, we extract the term frequency-based features $\bar{\mu}$ from the development set (***Dev Features***) and the test set (***Test Features***). Next, we generate translations for the development set using the pre-trained model and extract features from these translations (***Pretrained Features***). Using these three types of features, we compute the cosine similarity between two feature sets in the following two cases. **Case (i)** compares *Dev Features* and *Pretrained Features* to examine how much CDPG can improve over PRE-TRAINED. **Case (ii)** compares *Dev Features* and *Test Features* to indicate how uniformly the lexical distributions are aligned across the sampled sets. To account for differences in feature set lengths, which arise from zero counts due to mismatches in word usage, we compute similarity using both the *intersection* and the *union* of the sets.

Table 4 presents the analysis of features to complement Tables 1 and 2 in some representative domains. Focusing on *IT* of en→de, we observe that in Case (i), the similarity *sim.uni* is low (65.09), suggesting that the pre-trained model does not perform well on this domain. Consequently, CDPG successfully improves overall performance by aligning features more closely with those of the *IT* domain. Similar trends are also observed in the *Thesis* domain of en→zh and the *Education* domain of zh→en. In contrast, in the *Koran* domain, the similarity is already high (95.69), so the impact of CDPG is limited. A similar trend is observed in de→en. Turning to *Laws* of en→zh and *Science* of zh→en, although the similarity in Case (i) is relatively low, CDPG-based methods show only limited improvement. Looking at Case (ii), the similarity *sim.uni* is also quite low, indicating that the lexical distribution is scattered. In such cases, simply approximating the feature space is insufficient to improve BLEU or other metrics, as the issue lies in a distribution mismatch within the same domain data. However, even in these scenarios, both NIST and

Table 6: Translation examples from the outputs of Pre-Trained and CDPG. We select one short and one long sentence for each of en→de and en→zh. In #Changes, the numerator indicates the number of sentences in which CDPG made changes compared to Pre-Trained, and the denominator indicates the size of the test set. Words highlighted in red indicate correct domain-specific usage, those in blue represent updated terms that do not match the correct target term, and underlined words indicate inaccurate translations.

| Domain: IT | Pair: en→de | #Changes: 662/2000 |
|---|---|---|
| Input | SubDialog has one state, default. | |
| Reference | SubDialog hat nur einen Status, Standard. | |
| Pre-trained | SubDialog hat einen Zustand, default. | |
| CDPG | SubDialog hat einen Zustand, Standard. | |

| Domain: Medical | Pair: en→de | #Changes: 748/2000 |
|---|---|---|
| Input | 4 ml of solution in a 5 ml vial (type I glass) closed with a latex-free stopper (bromobutyl/ isoprene polymer) and a seal (lacquered plastic). | |
| Reference | 4 ml Lösung in einer 5 ml-Durchstechflasche (Glastyp I), die mit einem latexfreien Stopfen (Bromobutyl/Isoprenpolymer) und eine Kappe (lackierter Kunststoff) verschlossen ist. | |
| Pre-trained | 4 ml Lösung in einer 5-ml-Durchstechflasche (Glas Typ I), die mit einem latexfreien Stopfen (Brombutyl/Isoprenpolymer) und einem Siegel (Lackkunststoff) verschlossen ist. | |
| CDPG | 4 ml Lösung in einer 5 ml Durchstechflasche (Glas Typ I), die mit einem latexfreien Stopfen (Brombutyl/Isoprenpolymer) und einem Siegel (lackierter Kunststoff) verschlossen ist. | |

| Domain: Education | Pair: en→zh | #Changes: 408/790 |
|---|---|---|
| Input | What an absurd suggestion! | |
| Reference | 多荒谬的建议啊！ | |
| Pre-trained | 胡说八道！ | |
| CDPG | 多么荒谬的建议！ | |

| Domain: Thesis | Pair: en→zh | #Changes: 414/625 |
|---|---|---|
| Input | Newton's transformation family f w(z)=z-1wz w-1 containing only one complex parameter w(w≠0 or 1) is constructed from the transcendental mapping z→e z w+c. | |
| Reference | 用超越复映射F(z) =ezw+c构造出含有单参数w(w≠ 0或1)的牛顿变换族fw(z) =z- 1wzw-1模型，fw(z)有可数无穷多个极值点。 | |
| Pre-trained | 牛顿的变换型fw(z) =z-1wz W-1 仅包含一个复合参数w(w)0 或1) 的f(z) =z-1wz W-1。 | |
| CDPG | 牛顿的变换型fw(z) =z-1wz W-1 仅包含一个复合参数w(w)0 或1)，是用超常绘图ze z+c 构造的w-1模型。 | |

Confidence scores consistently improve, suggesting that CDPG still succeeds in generating domain-specific terms. In summary, CDPG is most effective when there exists a relatively large gap in feature distribution in the pre-trained model, indicating that it can achieve faithful domain shifts under such conditions.

Finally, to evaluate the sensitivity of CDPG to parameter settings, we fix the top-$p$ value at three levels: 0.5, 0.8, and 1.0. Table 5 shows the resulting score variations. Based on these results and the analysis in Table 4, we categorize the outcomes into two scenarios. First, when the similarity between *Dev Features* and *Pretrained Features* (Case i) is low and the similarity between *Dev Features* and *Test Features* (Case ii) is high, smaller top-$p$ values tend to yield better performance. This pattern is observed in domains such as *Thesis* of en→zh and *IT* of en→de. In contrast, using top-$p$ = 1.0 helps maintain performance in domains like *Laws* of en→zh and *Science* of zh→en. Second, when the similarity between *Dev Features* and *Pretrained Features* (Case i) is already high, the performance improvement from CDPG is generally limited. In such cases, score variation across different top-$p$ values is small, and top-$p$ = 1.0 tends to be the safest choice. In addition, we observe that the Confidence score consistently changes with the top-$p$ value, regardless of the actual translation quality. These results support our earlier discussion that the performance of CDPG is influenced by the properties of the provided monolingual lexical distribution features, such as the gap between the pre-trained model and the target domain, as well as the consistency of domain distributions.

## 6.2 Qualitative Analysis

Table 6 presents four representative translation instances that illustrate subtle effects of CDPG not fully captured by quantitative metrics. We first observe that CDPG only partially modifies the original model's knowledge, as demonstrated by conservative changes in the translations, and primarily enhances the model's

Table 7: Human evaluation results. In (i) Automatic Statistics, #test indicates the number of sentences in the test set, #diff denotes the number of sentences whose outputs differ between CDPG and Pre-Trained, change% is the proportion of changed outputs (#diff/#test), and win% is the percentage of sentences where CDPG obtains a BLEU score not lower than Pre-Trained. In (ii) Count by Feature, #term. represents the number of changes caused by terminology, #fit indicates how many of these changes match the provided monolingual features, term.% is the proportion of terminology-related changes (#term./#diff), and fit% is the percentage of terminology changes that correspond to target-domain features (#fit/#term.). In (iii) Human Judgment, #win, #lose, and #tie show human preferences for term selection among the #term. cases. win% indicates the proportion of clear improvements and win-tie% indicates the proportion of outputs that are judged as improved or at least reasonable under the conservative soft-constraint behavior of CDPG.

| | | (i) Automatic Statistics | | | | (ii) Count by Features | | | | (iii) Human Judgment | | | | |
|---|---|---|---|---|---|---|---|---|---|---|---|---|---|---|
| Pair | Domain | #test | #diff | change% | win% | #term. | #fit | term.% | fit% | #win | #tie | #lose | win% | win-tie% |
| en→zh | Education | 790 | 408 | 51.65 | 61.38 | 188 | 176 | 46.08 | 93.62 | 106 | 52 | 30 | 77.94 | 84.04 |
| en→zh | Thesis | 625 | 414 | 66.24 | 55.08 | 205 | 190 | 49.52 | 92.68 | 99 | 59 | 47 | 67.81 | 77.07 |
| zh→en | Laws | 456 | 276 | 60.53 | 61.83 | 142 | 140 | 51.45 | 98.60 | 89 | 14 | 39 | 69.53 | 72.54 |
| de→en | IT | 2,000 | 259 | 12.95 | 57.00 | 147 | 145 | 56.76 | 98.64 | 67 | 43 | 37 | 64.42 | 74.83 |
| en→zh | Laws | 456 | 268 | 58.77 | 44.40 | 107 | 102 | 38.06 | 95.33 | 30 | 51 | 26 | 53.57 | 75.70 |
| de→en | Medical | 2,000 | 215 | 10.75 | 48.47 | 64 | 59 | 29.77 | 92.19 | 24 | 32 | 8 | 75.00 | 87.50 |

term choice. Specifically, for the two en→de instances, regardless of sentence length, only domain-specific terms are modified without affecting the overall semantics or syntax, resulting in conservative behavior where not all test set inferences are altered. These findings confirm our motivation that CDPG can modify model knowledge harmlessly, avoiding issues such as catastrophic forgetting. Notably, they also explain the non-significant differences in BERTScore observed in Tables 1 and 2, since representation-level evaluation methods are not sensitive to word-level changes. In contrast, the consistent improvements in NIST scores highlight CDPG's effectiveness in conservatively enhancing domain-specific term usage.

However, these findings do not imply that CDPG benefits only word selection. In *Thesis* of en→zh, Pre-trained exhibits issues such as semantic loss and repetitive generation, whereas CDPG complements the missing semantics and mitigates the repetition. This improvement may be attributed to the enhanced confidence provided by GDC. Similarly, in the short sentence from en→zh, Pre-trained tends to translate the source into Chinese idioms that do not fully align semantically with the original sentence, i.e., ignoring the semantics of the word "suggestion". In contrast, CDPG correctly translates the key terms, suggesting that GDC enhances the model's focus on important words. In addition, in the long sentence of en→zh, the blue words represent an error in translation. This occurs because CDPG translates "transcendental" and "mapping" separately, as both words are present in the given features. This suggests that, since CDPG acts as a soft constraint, its use of keywords is not strictly enforced, but applied in a conservative manner.

## 6.3 Human Evaluation for Fine-grained Term-level Analysis

In Section 6.2, we found that CDPG makes conservative changes to translations and primarily enhances the model's term choice. Because the use of keywords is not strictly enforced but applied in a conservative manner, it effectively acts as a soft constraint, indicating that CDPG does not introduce destructive or harmful changes. Moreover, based on the discussion in Section 6.1, it is suggested that CDPG is the most effective when a relatively large gap exists in feature distribution in the pre-trained model, indicating that it can achieve faithful domain shifts under such conditions. From these observations, we hypothesize that CDPG-based methods function by encouraging the model to make term-level improvements according to the provided feature distributions. To verify this, we conduct human evaluations to analyze the model's behavior from a term-level perspective.

This evaluation consists of three steps: (i) a sentence-level analysis using automatic statistics, i.e., BLEU, to measure improvement rates in order to focus only on examples whose outputs were changed by CDPG; (ii) a manual examination of sentences where changes were caused by term substitutions, checking whether the

substituted terms appear in the target-domain features; (iii) finally, for cases where the substituted terms match the features, we measure the win rate via human evaluation to assess whether CDPG consistently produces faithful domain shifts at the term level. Table 7 shows the human evaluation results on six representative cases, including four cases where automatic statistics show substantial improvement, and two cases where the BLEU improvement is relatively small between CDPG and PRE-TRAINED in Tables 1 and 2.

First, from the results of (i) Automatic Statistics, we observe that the impact of the soft constraint of CDPG on outputs (change%) is around 50 to 60% for Chinese-related domains and slightly above 10% for German-related domains. This suggests that the extent of changes depends on how the test sets are constructed. Even when the number of changed outputs is small, for example 12.95% for de→en IT, the win rate, i.e., how often CDPG outperformed PRE-TRAINED among changed samples, still shows a positive trend. In contrast, for domains such as en→zh Laws, where the change ratio is high (58.77%), the final win rate is more limited (44.40%). This suggests that, under a conservative soft-constraint domain shift, the impact is determined primarily by which terms are substituted, rather than by the number of substitutions.

Next, based on the analysis in Section 6.1, i.e., the effectiveness of CDPG depends on the alignment between the provided features and the test domain, we further conduct (ii) Count by Features. We manually examine sentences whose changes come from term substitutions and check whether the substituted terms appear in our features. The results of fit% show that most terminology-related substitutions indeed correspond to features in the target domain, while a few mismatches can be attributed to the soft-constraint behavior of CDPG discussed in Section 6.2. This confirms that the improvements arise from faithful domain-specific terminology shifts guided by the provided feature distribution.

Finally, we conduct human evaluation in (iii) Human Judgment to assess whether the substitutions are reasonable for sentences involving terminology changes (#term.). Sentence-level comparisons between CDPG and PRE-TRAINED show that CDPG is preferred in a majority of cases in win%, and when ties are included (win-tie%), the preference reaches approximately 75%. This indicates that CDPG generally produces faithful yet harmless modifications, consistent with its soft-constraint nature. Taken together, these findings demonstrate that CDPG induces subtle terminology-level adjustments that may not be fully captured by automatic metrics such as BLEU, but remain reasonable and domain-appropriate. Therefore, we conclude that CDPG aligns with our motivation, as it performs a soft-constraint domain shift guided by the provided feature distribution and shows stable effectiveness through manual validation.

## 6.4 Will Other Domains Be Influenced?

The primary goal of CDPG is to encourage the distribution of the pre-trained model to align with the expectations of the given features. However, since CDPG fits the model to a single domain, there is a potential risk of reduced generalization to other domains. Therefore, as shown in Table 8, we conduct experiments to measure the performance changes of DYNAMIC CDPG across domains from two perspectives: (1) the relative difference between FINE-TUNED and DYNAMIC CDPG in the experimented domains; and (2) the change in performance of both FINE-TUNED and DYNAMIC CDPG on a generic domain.

First, FINE-TUNED consistently shows a decrease in both confidence and performance on the generic domain, whereas DYNAMIC CDPG achieves a significant increase in confidence in most cases, albeit with some fluctuations in performance. This indicates that the improvements achieved by our method generalize well, likely due to its intentionally conservative updates. While DYNAMIC CDPG generally demonstrates better generalization than FINE-TUNED, there are two types of exceptions: (1) Changes in confidence can affect generalization, as CDPG induces a global increase in confidence rather than domain-specific gains. However, this indirect influence is generally limited. For example, although the largest BLEU score drop caused by increased confidence is 1.13 on *Koran* of en→de, DYNAMIC CDPG achieves a 2.86 BLEU improvement on *IT*, which is significantly better than FINE-TUNED. (2) Performance in aligned cases is lower than in some cross-domain settings, such as *Thesis* of en→zh and *Medical* of en→de, suggesting that the provided *dev features* can have a negative impact. These results once again corroborate our analysis in Section 6.1, showing that the effectiveness of CDPG is closely tied to the quality and nature of the provided features.

Table 8: Relative differences between the scores of Fine-tuned and Dynamic CDPG. The columns and rows indicate the domains used for training and testing, respectively. Underlined values denote aligned cases where the training and testing domains are the same. *Gen.f.t.* and *Gen.d.c.* indicate the differences between Pre-trained and Fine-tuned, and between Pre-trained and Dynamic CDPG, respectively, on a generic domain (newstest2020). These serve as pivots for measuring relative differences.

| | | Confidence | | | | | BLEU | | | | |
|---|---|---|---|---|---|---|---|---|---|---|---|
| | | Education | Thesis | Science | Gen.f.t. | Gen.d.c. | Education | Thesis | Science | Gen.f.t. | Gen.d.c. |
| →zh | Education | 7.94 | 8.29 | 9.21 | -1.16 | 8.52 | 1.09 | 0.29 | -0.44 | -0.76 | -0.17 |
| | Thesis | 6.70 | 3.99 | 5.58 | -0.31 | 7.69 | 0.87 | 0.20 | 0.37 | -0.28 | 0.13 |
| | Science | 4.83 | 4.20 | 5.38 | -0.68 | 4.92 | 0.87 | 0.50 | 0.28 | -0.02 | 0.33 |
| →en | Education | 7.39 | 7.86 | 7.83 | -0.59 | 8.31 | 0.69 | -0.07 | -0.27 | -0.11 | 0.19 |
| | Thesis | 7.72 | 8.46 | 8.03 | -0.51 | 8.84 | 0.66 | -0.11 | 0.09 | -0.04 | 0.20 |
| | Science | 7.81 | 8.51 | 8.07 | -0.55 | 8.89 | 0.64 | -0.26 | -0.02 | -0.07 | 0.26 |
| | | IT | Medical | Koran | Gen.f.t | Gen.d.c. | IT | Medical | Koran | Gen.f.t | Gen.d.c. |
| →de | IT | 10.61 | 8.97 | 9.90 | -0.22 | 12.75 | 2.86 | 0.38 | -1.13 | -0.15 | -1.65 |
| | Medical | 8.32 | 6.61 | 7.11 | -0.27 | 9.47 | 2.44 | 0.28 | -0.63 | -0.07 | -0.90 |
| | Koran | 0.65 | 0.47 | -0.09 | -0.21 | -0.86 | 1.05 | 0.93 | -0.01 | -0.18 | -0.08 |
| →en | IT | 5.89 | 6.17 | 6.76 | -0.22 | 10.38 | 2.72 | -0.78 | -0.04 | -0.11 | -0.81 |
| | Medical | -1.02 | -0.05 | -0.82 | -0.29 | -1.50 | -0.65 | -0.42 | -0.11 | -0.06 | -0.18 |
| | Koran | 6.07 | 5.92 | 5.95 | -0.20 | 8.28 | 0.97 | -0.91 | 0.13 | -0.14 | -0.40 |

Table 9: Scores from experiments with mixed-domain data from two domains. The *main domain* data is fixed, and sentences from a *mixed domain* are added during feature construction. The model is then evaluated only on the main domain data. #Sent. indicates the number of added sentences from the mixed domain. The best value for each domain block (across different #Sent. settings) is shown in **bold**.

**English→German (Main: *IT*, Mix: *Medical*)**

| Method | #Sent. | Conf. | BLEU | P | R | F1 |
|---|---|---|---|---|---|---|
| Fine-tuned | 0 | 67.91 | **27.92** | **87.38** | **87.60** | **87.42** |
| | 500 | 67.82 | 27.63 | 87.35 | 87.57 | 87.39 |
| | 1,000 | 67.75 | 27.61 | 87.36 | 87.58 | 87.40 |
| | 2,000 | 67.61 | 27.27 | 87.32 | 87.55 | 87.36 |
| CDPG | 0 | 74.44 | 29.01 | 87.68 | 87.77 | 87.67 |
| | 500 | 74.24 | **29.70** | **87.70** | **87.80** | **87.70** |
| | 1,000 | 74.22 | 28.83 | 87.63 | 87.77 | 87.64 |
| | 2,000 | 74.14 | 29.43 | 87.68 | 87.79 | 87.68 |

**English→Chinese (Main: *Thesis*, Mix: *Laws*)**

| Method | #Sent. | Conf. | BLEU | P | R | F1 |
|---|---|---|---|---|---|---|
| Fine-tuned | 0 | 47.23 | **19.94** | 76.42 | **75.75** | **75.99** |
| | 750 | 47.14 | 19.77 | **76.44** | 75.73 | 75.98 |
| | 1,500 | 47.05 | 19.63 | 76.42 | **75.75** | 75.98 |
| | 3,000 | 46.83 | 19.13 | 76.37 | 75.74 | 75.95 |
| CDPG | 0 | 54.19 | 19.94 | 76.11 | 75.53 | 75.72 |
| | 750 | 54.16 | 20.06 | **76.25** | **75.59** | **75.81** |
| | 1,500 | 54.12 | **20.15** | 76.21 | **75.59** | 75.80 |
| | 3,000 | 54.01 | 20.10 | 76.24 | 75.58 | 75.80 |

Finally, we simulate a more realistic setting where the prepared data is not perfectly clean and may include content outside the target domain. We demonstrate the robustness of CDPG by comparing its performance trends to those of Fine-tuned in mixed-domain scenarios, where an extra domain dataset is intentionally introduced during tuning as contamination. Table 9 shows the results under several contamination settings. The results reveal that the performance of Fine-tuned consistently declines as the degree of domain mixing increases. In contrast, the performance of CDPG remains stable regardless of the domain mixture.

These findings suggest that, due to its conservative tuning approach, CDPG is less prone to issues such as catastrophic forgetting and overfitting. Moreover, even under more realistic conditions involving data contamination, CDPG achieves robust domain adaptation using only a small amount of monolingual data and target-side lexical distributions, outperforming typical fine-tuning methods that rely solely on parallel data.

Table 10: Results of CDPG with fixed top-$p$ values of 0.5, 0.8, and 1.0, evaluated over eight different random seeds, reporting the mean $\mu$ and standard deviation $\sigma$ as $\mu_{\pm\sigma}$. The row order follows Tables 4 and 5. The experimental settings and abbreviations are consistent with those in Table 5.

| Pair | Domain | top-$p = 0.5$ | | | top-$p = 0.8$ | | | top-$p = 1.0$ | | |
|---|---|---|---|---|---|---|---|---|---|---|
| | | Conf. | BLEU | F1 | Conf. | BLEU | F1 | Conf. | BLEU | F1 |
| en→de | IT | $74.42_{\pm 0.05}$ | $29.01_{\pm 0.26}$ | $87.66_{\pm 0.02}$ | $74.57_{\pm 0.10}$ | $29.00_{\pm 0.22}$ | $87.66_{\pm 0.02}$ | $67.64_{\pm 0.21}$ | $28.03_{\pm 0.34}$ | $87.40_{\pm 0.06}$ |
| en→de | Koran | $66.97_{\pm 0.04}$ | $18.47_{\pm 0.07}$ | $80.73_{\pm 0.02}$ | $67.07_{\pm 0.05}$ | $18.60_{\pm 0.07}$ | $80.75_{\pm 0.02}$ | $60.92_{\pm 0.21}$ | $18.91_{\pm 0.08}$ | $80.87_{\pm 0.02}$ |
| de→en | Law | $78.11_{\pm 0.04}$ | $51.60_{\pm 0.02}$ | $95.86_{\pm 0.01}$ | $78.31_{\pm 0.05}$ | $51.55_{\pm 0.03}$ | $95.86_{\pm 0.00}$ | $71.65_{\pm 0.22}$ | $51.62_{\pm 0.04}$ | $95.89_{\pm 0.01}$ |
| de→en | Medical | $82.79_{\pm 0.03}$ | $44.57_{\pm 0.04}$ | $96.54_{\pm 0.00}$ | $83.07_{\pm 0.07}$ | $44.86_{\pm 0.04}$ | $96.54_{\pm 0.00}$ | $77.65_{\pm 0.21}$ | $45.31_{\pm 0.30}$ | $96.54_{\pm 0.01}$ |
| en→zh | Thesis | $54.26_{\pm 0.11}$ | $19.98_{\pm 0.06}$ | $75.81_{\pm 0.06}$ | $54.14_{\pm 0.13}$ | $19.96_{\pm 0.08}$ | $75.84_{\pm 0.06}$ | $47.14_{\pm 0.20}$ | $19.96_{\pm 0.06}$ | $75.74_{\pm 0.06}$ |
| en→zh | Laws | $68.57_{\pm 0.07}$ | $50.81_{\pm 0.15}$ | $89.59_{\pm 0.03}$ | $68.88_{\pm 0.12}$ | $51.09_{\pm 0.17}$ | $89.59_{\pm 0.03}$ | $61.80_{\pm 0.12}$ | $51.85_{\pm 0.10}$ | $89.71_{\pm 0.03}$ |
| zh→en | Education | $66.12_{\pm 0.06}$ | $23.82_{\pm 0.18}$ | $94.41_{\pm 0.01}$ | $66.19_{\pm 0.08}$ | $23.90_{\pm 0.15}$ | $94.38_{\pm 0.01}$ | $59.85_{\pm 0.20}$ | $23.68_{\pm 0.12}$ | $94.33_{\pm 0.02}$ |
| zh→en | Science | $64.12_{\pm 0.05}$ | $16.02_{\pm 0.13}$ | $92.69_{\pm 0.01}$ | $64.05_{\pm 0.10}$ | $16.09_{\pm 0.04}$ | $92.39_{\pm 0.01}$ | $57.69_{\pm 0.17}$ | $16.30_{\pm 0.08}$ | $92.68_{\pm 0.02}$ |

## 6.5 Robustness Analysis of CDPG to Randomness

Finally, in our previous experiments, e.g., Sections 5 and 6, we mainly reported single-run results, e.g., Tables 1, 2, and 5, to enable straightforward analysis and discussion based on consistent results. To evaluate the robustness of CDPG, we additionally conducted multiple runs with different random seeds and report the mean $\mu$ and standard deviation $\sigma$. Table 10 presents the results obtained from eight different seed values, while keeping all other settings and hyperparameters the same as in Table 5.

From Table 10, we observe that the standard deviations are small, indicating that CDPG is stable. In fact, the coefficient of variation ($\sigma/\mu$) is at most 1.21%, and in most cases less than 1%. In the CDPG-based method, the only component largely influenced by the random seed is the step in which the target language model generates sentences $\boldsymbol{x}$ via nucleus sampling to train the distribution feature $\boldsymbol{\lambda}$ for the EBMs. Since the sampling is conducted from the entire distribution $\mathcal{X}$ of the target language model, a sufficient number of samples will closely approximate the true distribution, resulting in stable performance regardless of the random seed value. On the other hand, as discussed in Section 6.1, varying the top-$p$ value changes the underlying sampling distribution, introducing fluctuations in diversity and, consequently, degradations in the results. Even with many samples, such diversity variation inherently affects the outcomes, suggesting the necessity of controlling the diversity parameter. This further supports the rationale behind methods such as DYNAMIC CDPG, which explicitly adapt the sampling parameters to account for such variation.

In summary, when sentences $\boldsymbol{x}$ are sampled from the same distribution $\mathcal{X}$, the method is largely insensitive to randomness and remains robust. To control or influence the behavior of the system, it is therefore essential to adjust the sampling method or diversity-related parameters that directly modify the underlying distribution.

## 7 Limitations and Future Extensions

We conduct a comprehensive analysis; however, we acknowledge three main limitations in this work.

First, as stated in Sections 5 and 6.2, representation-level MT evaluation methods are not sensitive to the improvements made by CDPG, which results in only minor differences in BERTScore in particular. Moreover, although NIST provides a more reasonable assessment of domain-specific terminology and aligns more closely with our objectives, it remains limited by its BLEU-style surface-level representation design. Therefore, exploring how semantic-level evaluation methods can better capture word-specific changes remains an important direction for future work. Additionally, we did not include modern neural fine-tuned metrics, such as COMET (Rei et al., 2020) and BLEURT (Sellam et al., 2020), as part of our main evaluation. These metrics are fine-tuned on human-generated MT quality annotation data (Ma et al., 2019), but such data often fails to capture subtle patterns such as named entity differences (Amrhein & Sennrich, 2022; Glushkova et al., 2023). Moreover, due to overfitting to the annotation data, these metrics tend to favor outputs that are

closer to the in-domain data of their fine-tuning sets (Zouhar et al., 2024a;b). Consequently, we determined that such fine-tuned metrics are not suitable for evaluating domain adaptation experiments.

Second, our primary goal is to demonstrate a novel practical application of massive EBMs using CDPG in a downstream task, with domain adaptation for machine translation as a case study. To maintain a clear focus on this objective, we adopted simple encoder-decoder architecture NMT models (Tiedemann & Thottingal, 2020). We did not employ multilingual large models such as NLLB-200 (Team et al., 2022) or M2M100 (Fan et al., 2021), as they introduce additional complexity, including translation issues arising from multilingualism. Furthermore, extending our approach to translation-specific decoder-only large language models (Alves et al., 2024; Xu et al., 2025; Feng et al., 2025; Cui et al., 2025) or applying it to other domain datasets (Oncevay et al., 2025; Zhang et al., 2025; Cui et al., 2025) was not feasible given our available computational resources and would have made it harder to isolate the core effects of our method. Nonetheless, we believe our experiments already provide a comprehensive demonstration of the method's applicability and sufficiently address our research question. While such extensions remain promising future directions, we believe this work is a valuable pioneering study that lays the groundwork for broader applications.

Third, in this paper, DYNAMIC CDPG is described primarily using nucleus sampling. However, the method is not restricted to this choice and can also be applied to other sampling strategies (Fan et al., 2018; Vijayakumar et al., 2018; Hewitt et al., 2022; Minh et al., 2025). The core mechanism lies in adjusting sampling diversity, and nucleus sampling was adopted merely to maintain consistency with prior work (Khalifa et al., 2021). The primary focus of this study is domain adaptation for neural machine translation in low-resource settings. Our key finding is that the selection of sampling diversity during EBM approximation substantially affects performance, and dynamically adjusting this diversity is effective. Therefore, exploring alternative parameterized sampling strategies and identifying optimal configurations constitutes a natural direction for future work. Moreover, we primarily used BLEU to monitor the tuning process, although other evaluation metrics could also be employed. Even though BLEU was used for monitoring, the results in Sections 5 and 6 show consistent improvements in other metrics such as BERTScore, suggesting that the approach generalizes beyond a specific metric. Therefore, we expect similar behavior when using alternative evaluation metrics. Potential future extensions include developing more suitable metrics for guiding CDPG or exploring monitoring strategies.

## 8 Related Work

When using parallel data, Luong & Manning (2015); Freitag & Al-Onaizan (2016) perform domain adaptation by first training on large-scale general-domain data, then fine-tuning on a small amount of in-domain data. Chu et al. (2017) instead mix general and in-domain data for training at once. Further efficiency in domain adaptation has been pursued through techniques such as adding domain tags (Kobus et al., 2017; Britz et al., 2017), subword-aware tokenization (Enomoto et al., 2023), and training data sampling (Wang et al., 2017). However, direct fine-tuning with a small amount of data often leads to overfitting, prompting proposals of knowledge distillation (Dakwale & Monz, 2017) and regularization strategies (Miceli Barone et al., 2017).

In the context of monolingual data utilization, several methods have been explored such as back translation (Sennrich et al., 2016), direct learning from monolingual data as LM (Gulcehre et al., 2015; Zhang & Zong, 2016; Domhan & Hieber, 2017; Burlot & Yvon, 2018), exploiting task-specific features (Dou et al., 2019b;a), utilizing knowledge graphs (Moussallem et al., 2019; Zhao et al., 2020; Costa et al., 2022; Conia et al., 2024), and nearest neighbor search (Farajian et al., 2017; Bapna & Firat, 2019; Zheng et al., 2021; Khandelwal et al., 2021; Wang et al., 2022; Deguchi et al., 2023; Agrawal et al., 2023), and the combination of unsupervised NMT methods and back-translation technique (Mahdieh et al., 2020; Zhang et al., 2022). However, these approaches usually require large-scale monolingual data, which is not always available in specialized domains. Therefore, they may not be suitable for the extremely low-resource domain adaptation settings.

For terminology-constrained decoding, hard-constrained decoding methods (Hokamp & Liu, 2017; Post & Vilar, 2018; Hu et al., 2019; Post et al., 2019), which force the model to decode specific terminology, and soft-constrained decoding methods (Song et al., 2019; Chen et al., 2020), which apply post-editing techniques using phrase tables, have been proposed. However, since these approaches require predefined constrained vocabularies, they face challenges in real NMT scenarios that demand inductive domain adaptation.

CDPG (Korbak et al., 2022), which our study builds upon, focused only on minor input perturbations, e.g., replacing digits with spelled-out numbers, and did not address large-scale domain adaptation based on the full target-domain distribution. Regarding reinforcement learning methods (Ranzato et al., 2016; Kreutzer et al., 2017; Choshen et al., 2020; Yang et al., 2024), outside of the GDC framework, rewards are based only on overall scores such as BLEU, without the ability to impose fine-grained constraints. Furthermore, there is a potential for causing catastrophic forgetting, making scaling like in this study particularly challenging.

## 9 Conclusion

In this study, we explored whether unsupervised domain adaptation in machine translation can be effectively performed using only naive statistics, i.e., the monolingual lexical distribution, which can be easily obtained from target-side domain data without requiring bilingual supervision. We demonstrated this by imposing large-scale distributional constraints through EBMs at scale using the CDPG method, leveraging features derived from the entire target-domain corpus, and confirmed its effectiveness across multiple domains and language directions. This work is the first to demonstrate the practical applicability of CDPG of the GDC framework to a realistic downstream task, using the unsupervised domain adaptation of pre-trained NMT models as a case study. Although this work focused on word-level lexical distributions as the guiding signal, we believe that future work should explore alternative or complementary feature representations, such as $n$-gram statistics or language model embeddings, to further enhance fine-tuning within the GDC framework.

## Acknowledgements

We sincerely thank the anonymous reviewers and the action editor for their valuable comments and suggestions. Their constructive feedback significantly strengthened this work and encouraged its publication. We are deeply grateful for their recognition of our contributions and their support toward publication.

We also extend our sincere thanks to Sei Iwata, an early adopter of this project. This work is based on extensive discussions and iterative trial-and-error with him. We are pleased to be able to inform him that these efforts have ultimately led to publication.

Regarding the author contributions of this paper, Yusuke Sakai and Zhi Qu contributed equally to this work. Yusuke Sakai led the project from the initial commitment through final acceptance and was responsible for the main manuscript writing. Zhi Qu contributed primarily to code preparation and extensive experiments, drafted the Results and Discussion sections, and joined the project after the departure of Sei Iwata, engaging in extensive discussions thereafter.

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
