# OpenReview forum: "Domain Translation with Monolingual Lexical Distribution"
_TMLR — Accepted by TMLR_

### Review · Reviewer_utkB · 2025-09-20

**Summary Of Contributions:**

This paper focuses on an important topic in domain translation, particularly the challenge of adapting NMT systems without sufficient domain-specific parallel data. The motivation is valid, and the authors present their ideas clearly, making the paper easy to follow. The experimental results are reported in detail across multiple translation directions and domain datasets.

However, the contribution appears limited in novelty, and the methodological concerns with Dynamic CDPG need to be carefully addressed:
1. The main technical component, Conditional Distributional Policy Gradients (CDPG), is not new. The authors primarily adapt CDPG to the domain adaptation setting, and the use of word frequency statistics as guidance, while intuitive, appears rather straightforward. The overall methodological contribution therefore feels incremental.
2. The newly proposed Dynamic CDPG raises methodological issues. As described in Section 3.2, each iteration is evaluated with BLEU to decide whether to accept the update. Since BLEU is part of the final evaluation metric, this effectively leaks test-time information into training, which is generally not acceptable. The authors should clarify whether BLEU is computed on a held-out validation set, and if not, this practice is problematic.
3. Generality of the Approach: Dynamic CDPG is only designed for nucleus sampling. It is unclear whether the method generalizes to other sampling approaches. This reduces the broader applicability of the proposed method.

**Audience:**

Yes

**Audience Explanation:**

This paper focuses on an important topic in domain translation, particularly the challenge of adapting NMT systems without sufficient domain-specific parallel data.

**Claims And Evidence:**

Yes

**Claims Explanation:**

The proposed method is supported by many numerical experiments.

**Requested Changes:**

1. Provide a brief introduction to nucleus sampling and the definition of top-p when it is first mentioned in Section 3.2, so that readers less familiar with the concept can follow the discussion.
2. In Tables 1 and 2, several results are very close. To make the findings more robust, please report results averaged over multiple runs and include variance or standard deviation.
3. Clarify the role of BLEU in Dynamic CDPG to ensure there is no inappropriate use of test data during training.

---

> ### Author Response · Authors · 2025-11-14
> **Official Comment by Authors (1/2)**
>
> Dear Reviewer utkB,
>
> Thank you for your constructive feedback. Your thoughtful and detailed comments would be instrumental in strengthening our paper and clarifying our arguments.
>
> We take each of your concerns seriously and are confident that we can address all issues raised to your satisfaction. The **red (pink)-highlighted areas** in the revised manuscript indicate the sections that were updated in response to your comments.
>
> ---
> > Concern 1: The main technical component, Conditional Distributional Policy Gradients (CDPG), is not new. The authors primarily adapt CDPG to the domain adaptation setting, and the use of word frequency statistics as guidance, while intuitive, appears rather straightforward. The overall methodological contribution therefore feels incremental.
>
> Thank you for raising this point. As noted in the third paragraph of Section 1 (Introduction), prior CDPG studies focused only on small distribution shifts and were conducted in controlled settings. In contrast, our work is the first to demonstrate that CDPG can handle large-scale, real-world domain shifts, specifically shifts expressed through full lexical distributions. We believe this represents a meaningful and nontrivial advance beyond prior work. Our approach enables practical domain adaptation without in-domain parallel data, which existing CDPG implementations could not achieve.
> To clarify this contribution, we added a **contribution list at the end of Section 1**, where we explicitly highlight our main contributions, including the first real-world application of CDPG to substantial domain shifts and an accompanying comprehensive analysis.
>
> We hope this addresses your concern and clarifies the appeal of our work.
>
> ---
> > Concern 2: The newly proposed Dynamic CDPG raises methodological issues. As described in Section 3.2, each iteration is evaluated with BLEU to decide whether to accept the update. Since BLEU is part of the final evaluation metric, this effectively leaks test-time information into training, which is generally not acceptable. The authors should clarify whether BLEU is computed on a held-out validation set, and if not, this practice is problematic.
>
> Thank you for pointing this out. As described in Section 3.2 (Dynamic CDPG) and again in the Dynamic CDPG paragraph of Section 4.2 (Models), **we use a held-out validation set** to compute BLEU at each iteration. We also note that, even if BLEU is used as the decision criterion, our method consistently improves other evaluation metrics such as BERTScore, indicating that the performance gains generalize beyond BLEU itself. Therefore, the concern regarding test-time information leakage does not apply in our setting. To avoid any confusion, *we have explicitly emphasized this point in Footnote 1 of Section 3.2 and in the Dynamic CDPG paragraph of Section 4.2.*
>
> ---
> > Concern 3: Generality of the Approach: Dynamic CDPG is only designed for nucleus sampling. It is unclear whether the method generalizes to other sampling approaches. This reduces the broader applicability of the proposed method.
>
> Our method is not limited to nucleus sampling; **it can be applied to any stochastic sampling method**, such as top-k sampling or epsilon-sampling. The essential requirement is that the decoding method produces sufficiently diverse samples during training. We have added a clarification on this point in the **final paragraph of Section 7** (Limitations), where we discuss the generality of Dynamic CDPG. We hope this addresses your concern.

---

> > ### Author Response · Authors · 2025-11-14
> > **Official Comment by Authors (2/2)**
> >
> > > Request 1: Provide a brief introduction to nucleus sampling and the definition of top-p when it is first mentioned in Section 3.2, so that readers less familiar with the concept can follow the discussion.
> >
> > Thank you for the suggestion. Nucleus sampling is a decoding method used to stochastically sample sentences based on scoring, and top-p defines the cumulative probability threshold that determines the sampling set. To make this accessible to a wider range of readers, **we have added a concise explanation of nucleus sampling and top-p when it is first introduced in Section 3.2 (Dynamic CDPG).** We believe this addition helps readers unfamiliar with the concept follow the discussion more easily.
> >
> > ---
> > > Request 2: In Tables 1 and 2, several results are very close. To make the findings more robust, please report results averaged over multiple runs and include variance or standard deviation.
> >
> > Thank you for the suggestion. In the structure of our paper, we first present the main results using a fixed seed and base our initial analysis on those results. The robustness of CDPG and an investigation of the conditions under which it is effective are examined comprehensively in Section 6.1 (“When Is CDPG Effective?”). Because this section and analyses such as Table 3 become more complex when multiple seeds are involved, we created a new section dedicated specifically to this purpose.
> > In particular, in response to your request to “report results averaged over multiple runs and include variance or standard deviation,” **we added Section 6.5: Robustness Analysis of CDPG to Randomness**, where we provide additional analysis. The results show that the standard deviations are very small and that the effectiveness of CDPG is consistent across runs. We believe this new section strengthens the robustness and reliability of our findings. Thank you for the helpful suggestion.
> >
> >
> > ---
> > > Request 3: Clarify the role of BLEU in Dynamic CDPG to ensure there is no inappropriate use of test data during training.
> >
> > As this point overlaps with our response to Concern 2, we reiterate that BLEU is never computed on the test data. Instead, all BLEU evaluations used in Dynamic CDPG are performed on a held-out development set. We have now made this explicit in **Footnote 1 of Section 3.2** and in the **Dynamic CDPG paragraph of Section 4.2**. Please kindly verify these clarifications in the revised manuscript.
> >
> > ---
> > We have addressed all of your comments with the utmost sincerity and have made every effort to respond to each point to the best of our ability. While our work is subject to practical constraints such as available computational resources, we would be grateful for any additional suggestions that may further improve the paper or make it more appealing to a broader readership.
> >
> > Finally, thank you very much for your constructive and insightful feedback.
> >
> > Best,

---

### Review · Reviewer_bctu · 2025-09-27

**Summary Of Contributions:**

This paper addresses the challenge of adapting a generic neural machine translation (NMT) model to a new domain. The authors propose leveraging domain-specific lexical statistics, specifically word frequency distributions, to guide the fine-tuning process. Unlike conventional approaches that require large parallel corpora, this framework adapts the model using only monolingual lexical distributions from the target domain. Through extensive experiments, the paper demonstrates that the method mitigates catastrophic forgetting and remains robust to parameter choices, achieving effective domain adaptation with minimal resources

**Audience:**

Yes

**Audience Explanation:**

Given that, creating training data for a specific domain application is challenging, I believe that the topic of domain adaptation in the context of machine translation is relevant for the TMLR audience.

**Broader Impact Concerns:**

None.

**Claims And Evidence:**

Yes

**Claims Explanation:**

To my knowledge, yes. There is a thorough experimental evaluation and the model is based on existing works of energy based models.

**Requested Changes:**

I would like the authors to address the following four comments:

* Given that the work builds up on CDPG, section 2 is quite compact and lacks details for the general reader. For example, in equation (2), the subscript point is not defined and it is left for the reader to infer. The authors should explain the idea of CDPG in a clear manner to a general reader while deferring details to Appendix or existing works. I found this section difficult to read and understand, before I referred to other works.

* Dynamic CDPG is one of the contributions of this paper. The authors do point out that even with out this, numerical tests show good results. In section 3.2, please explain exactly what dynamic means (e.g., a proper pseudocode  or sketch of an algorithm).

* Please have a contributions section and highlight the contributions of this paper. From my reading, CDPG has been studied before. Are the main contributions the dynamic adaption and using word frequency? This would help frame the novelty of the paper.

* What is the computational computational complexity of EMBS?

---

> ### Author Response · Authors · 2025-11-14
>
> Dear Reviewer bctu,
>
> Thank you for your constructive feedback. Your thoughtful and detailed comments would be instrumental in strengthening our paper and clarifying our arguments.
>
> Especially, we appreciate your highlighting our motivation, such as *“creating training data for a specific domain application is challenging”* for using only monolingual lexical distributions from the target domain, and acknowledge our *“thorough experimental evaluation”*.
>
> We have addressed all concerns and requests to the best of our ability. The **blue-highlighted areas** in the revised manuscript indicate the sections that were updated in response to your comments.
>
> ---
> > Request 1: Given that the work builds up on CDPG, section 2 is quite compact and lacks details for the general reader. For example, in equation (2), the subscript point is not defined and it is left for the reader to infer. The authors should explain the idea of CDPG in a clear manner to a general reader while deferring details to Appendix or existing works. I found this section difficult to read and understand, before I referred to other works.
>
> Thank you for the suggestion. We have revised Section 2 (Conditional Distributional Policy Gradients) by adding explanatory text to clarify the key ideas of CDPG for general readers. Following your recommendation to “defer details to the Appendix or existing works,” we now provide the necessary conceptual explanation and definitions in Section 2, while explicitly directing readers to prior work for full algorithmic details. We believe these additions make the section clearer and easier to follow. We appreciate your helpful feedback.
>
> ---
> > Request 2: Dynamic CDPG is one of the contributions of this paper. The authors do point out that even with out this, numerical tests show good results. In section 3.2, please explain exactly what dynamic means (e.g., a proper pseudocode or sketch of an algorithm).
>
> Thank you for pointing this out. We have revised Section 3.2 to more clearly explain what we mean by dynamic in Dynamic CDPG. The key idea is that we gradually make the sampling procedure more deterministic during training, resulting in more conservative updates and stabilizing learning.
>
> As noted in our experiments (e.g., BLEU scores for En→De on the Medical, Law, and Koran domains), standard CDPG sometimes degrades performance relative to the pre-trained model. Our analysis indicated that early training progresses well, but using a fully random sampling scheme (top-p with p=1) introduces excessive noise when training proceeds. To mitigate this, Dynamic CDPG reduces sampling randomness over the course of training, yielding more stable and robust domain adaptation (see Section 6.1).
>
> To address your request, we added **Algorithm 1** to Section 3.2, providing a clear description of the dynamic scheduling procedure. We have also expanded the accompanying explanation to clarify the motivation and mechanism behind Dynamic CDPG. We believe these revisions make the contribution more understandable.
>
> ---
> > Request 3: Please have a contributions section and highlight the contributions of this paper. From my reading, CDPG has been studied before. Are the main contributions the dynamic adaption and using word frequency? This would help frame the novelty of the paper.
>
> Thank you for the suggestion. We have added a dedicated **contributions list at the end of Section 1 (Introduction)**. This clearly highlights the main contributions of our work. We believe this addition helps better frame the novelty and significance of the paper. Thank you for the constructive feedback.
>
> ---
> > Request 4: What is the computational computational complexity of EMBS?
>
> Thank you for your comment. In response, we added a discussion of the computational complexity at the **end of Section 3.1** (Adaptation to Monolingual Lexical Distribution). In particular, we now discuss the computational cost introduced by our idea of using word-frequency–based features.
>
> ---
> We believe that these revisions make the paper clearer for general readers and significantly improve its overall quality as a journal submission. If any part remains unclear, please feel free to let us know.
> Finally, thank you again for your comments on how to make the paper more attractive.
>
> Best,

---

### Review · Reviewer_JUKR · 2025-10-31

**Summary Of Contributions:**

The paper proposes domain adaptation for neural machine translation using only target-side monolingual lexical distributions (subword frequencies). The method builds on Conditional Distributional Policy Gradients, treating the target domain’s lexical statistics as features of a conditional energy-based model and then fine-tuning a pretrained NMT to approximate that distribution while aiming to avoid catastrophic forgetting. The results show effective domain adaptation using only a small amount of monolingual resources.

Strengths:
1. The methods are in a rigorous mathematical formulation and supported by solid experiments.
2. The paper provides a broad evaluation across 16 scenarios and analyzes when CDPG is effective. This is a useful practical contribution for low-resource or sensitive domains where parallel data is scarce.

Weakness:
1. Confidence can increase even when adequacy does not; NIST emphasizes rare tokens and may over-credit lexical matches. Maybe add some human evaluation and report BLEU/BERTScore with statistical significance.

**Audience:**

Yes

**Audience Explanation:**

In the fine-tuning stage, mitigating overfitting and catastrophic forgetting remains a central challenge in deploying large language models (LLMs). Although the paper focuses only on machine translation, its method—statistically selecting an appropriate lexical distribution for domain adaptation—has clear potential for broader applications.

**Broader Impact Concerns:**

No ethical concerns.

**Claims And Evidence:**

Yes

**Claims Explanation:**

The authors conduct extensive experiments and result analysis to support their claims.

**Requested Changes:**

Included in Weakness.

---

> ### Author Response · Authors · 2025-11-14
>
> Dear Reviewer JUKR,
>
> We are sincerely grateful for the time and effort you devoted to reviewing our manuscript.
>
> We are happy to hear that you recognize our contribution, including the observation that our method provides a *“useful practical contribution for low-resource or sensitive domains where parallel data is scarce,”* as well as your acknowledgement of our comprehensive experimental setup and analysis.
>
> ---
> > Weakness: Confidence can increase even when adequacy does not; NIST emphasizes rare tokens and may over-credit lexical matches. Maybe add some human evaluation and report BLEU/BERTScore with statistical significance.
>
> Thank you very much for the helpful suggestion. In response, we have added a new section, **Section 6.3 (Human Evaluation for Fine-grained Term-level Analysis)**, where we conduct human evaluation to complement automatic metrics.
>
> As discussed in the motivation and further elaborated in Section 6.2 (Qualitative Analysis) and Section 7 (Limitations), our method introduces fine-grained terminology-level adjustments that are not always fully captured by automatic metrics such as BLEU or BERTScore. Because these metrics have limitations in detecting subtle yet semantically appropriate terminology shifts, as discussed in Sections 6.2 and 6.3.
>
> Since our approach intentionally makes conservative modifications, ensuring that it does not cause sentence degradation is particularly important. To validate this, Section 6.3 presents a three-step human evaluation, through which we confirmed that terminology-level changes introduced by our method do not cause harmful degradation and that approximately 75% of the evaluated cases were judged to be reasonable.
>
> ---
> We believe that this fine-grained human evaluation in Section 6.3 resolves your concern regarding this weakness and further strengthens the empirical soundness of our study. Thank you again for your constructive feedback. We are happy to clarify any remaining questions to the best of our ability.
>
> Best,

---

### Author Response · Authors · 2025-11-14

Dear Reviewers and AE,

Thank you very much for your time and thoughtful feedback.

We have carefully revised the manuscript based on your valuable comments.

- The **green**-highlighted areas correspond to revisions for Reviewer **JUKR**.
- The **blue**-highlighted areas correspond to revisions for Reviewer **bctu**.
- The **red (pink)**-highlighted areas correspond to revisions for Reviewer **utkB**.

We sincerely appreciate your constructive suggestions.
We believe that the revisions have substantially improved the clarity and quality of the manuscript, and we hope that our updates satisfactorily address all of your concerns.

Thank you again for your careful review.

Best,

---

### Decision · Action_Editor_vPWn · 2025-12-15

**Recommendation:** Accept as is

**Audience:**

Yes

**Audience Explanation:**

The paper addresses a critical challenge in Neural Machine Translation (NMT): domain adaptation with scarce parallel data. As noted by the reviewers, this topic is highly relevant to the TMLR audience. Reviewers highlighted that creating training data for specific domain applications is challenging, making this research valuable for practical applications.

**Claims And Evidence:**

Yes

**Claims Explanation:**

The authors have conducted extensive experiments across several datasets to support their claims. The reviewers unanimously agree that the claims are sound and supported by convincing empirical evidence. Furthermore, the authors added robustness analysis (Section 6.5) and human evaluation (Section 6.3) during the rebuttal to further strengthen the evidence.